# Prognostic Nutritional Index and Instant Nutritional Assessement Are Associated with Clinical Outcomes in a Geriatric Cohort of Acutely Inpatients

**DOI:** 10.3390/nu16091359

**Published:** 2024-04-30

**Authors:** Cristiano Capurso, Aurelio Lo Buglio, Francesco Bellanti, Gianluigi Vendemiale

**Affiliations:** Department of Medical and Surgical Sciences, University of Foggia, Viale Luigi Pinto 1, 71122 Foggia, Italy; aurelio.lobuglio@unifg.it (A.L.B.); francesco.bellanti@unifg.it (F.B.); gianluigi.vendemiale@unifg.it (G.V.)

**Keywords:** malnutrition, elderly, mortality, albumin, lymphocytes, prognostic nutritional index, Instant Nutritional Index

## Abstract

Background: Among elderly inpatients, malnutrition is one of the most important predictive factors affecting length of stay (LOS), mortality, and risk of re-hospitalization. Methods: We conducted an observational, retrospective study on a cohort of 2206 acutely inpatients. Serum albumin and lymphocytes were evaluated. Instant Nutritional Assessment (INA) and the Prognostic Nutritional Index (PNI) were calculated to predict in-hospital mortality, LOS, and risk of rehospitalization. Results: An inverse relationship between LOS, serum albumin, and PNI were found. Deceased patients had lower albumin levels, lower PNI values, and third- and fourth-degree INA scores. An accurate predictor of mortality was PNI (AUC = 0.785) after ROC curve analysis; both lower PNI values (HR = 3.56) and third- and fourth-degree INA scores (HR = 3.12) could be independent risk factors for mortality during hospitalization after Cox regression analysis. Moreover, among 309 subjects with a lower PNI value or third- and fourth-class INA, hospitalization was re-hospitalization. Conclusions: PNI and INA are two simple and quick-to-calculate tools that can help in classifying the condition of hospitalized elderly patients also based on their nutritional status, or in assessing their mortality risk. A poor nutritional status at the time of discharge may represent an important risk factor for rehospitalization in the following thirty days. This study confirms the importance of evaluating nutritional status at the time of hospitalization, especially in older patients. This study also confirms the importance for adequate training of doctors and nurses regarding the importance of maintaining a good nutritional status as an integral part of the therapeutic process of hospitalization in acute departments.

## 1. Introduction

According to the World Health Organization (WHO), the term malnutrition refers to a broad group of conditions, including both undernutrition (wasting, stunting, and underweight), inadequate intake of vitamins or minerals, and overweight and obesity, with diet-related, non-communicable diseases as a result [1]. Older people in particular may be at increased risk due to physiological alterations in body composition during aging (e.g., the loss of skeletal muscle mass, ‘sarcopenia’, and associated muscle protein), and reduction in appetite (e.g., “anorexia of aging”) [2,3,4,5,6]. Elderly people, in particular frail elderly people, hospitalized in acute hospital departments already suffer from chronic diseases, with signs and symptoms of physical and cognitive deterioration, and in complex home therapies characterized by the intake of numerous drugs [7]. 

The geriatric patient is not characterized solely by advanced age, but rather by the presence of numerous chronic pathologies, together with the presence in many cases of the well-known geriatric syndrome which is frailty. By frailty, we mean the state of greater vulnerability, or reduced resilience, in response to a stressful event, which increases the risk of adverse outcomes, including falls, delirium, and disability [8,9,10,11,12,13]. In these subjects, a small complication (for example a new drug, a small infection, or a small surgery) causes a significant and disproportionate change in the state of health, i.e., from independent to dependent, from mobile to immobile, from postural stability to propensity for falling, or from lucid to delirious. Because of both the cumulative decline of physiological systems, which is not limited to physical functions but can also affect cognitive functions, and various socio-environmental factors, functional reserves and the capacity for resilience are reduced, inducing a state of greater vulnerability to even mild stressful events [14]. Malnutrition is therefore a high-risk condition for the development of frailty.

It is also now known that there is substantial variability in the aging process between men and women. In general, women live longer than men live, but are frailer in old age, with a consequent poorer state of health. On the contrary, men, despite having a reduced life expectancy compared to women, maintain good levels of physical functionality even at an advanced age, resulting in a better state of health and a lower risk of frailty. This statement is supported by evidence that has shown that men experience more rapid and earlier age-associated immunoinflammatory changes than women, and that these changes can be attributed to both hormonal and environmental factors [15].

In acutely ill patients, disease-related malnutrition may occur because of a catabolic state triggered by systemic inflammation secondary to a concomitant disease. The association of this condition with a negative energy balance impacts adversely on body associated in with depression of the immune system, sarcopenia, and increased incidence of complications (e.g., infections, pressure ulcers), as well as with longer hospital stays, higher readmission rates, greater health care costs, and increased hospital and long-term patient mortality [16,17,18,19]. 

Among inpatients, malnutrition at admission is one of the most important negative predictive factors affecting the risk of several clinical outcomes, such as length of stay, treatment outcome of the main disease and comorbidities, mortality, and complications. 

To date, a significant proportion of hospital patients show signs of malnutrition upon admission and continue to deteriorate nutritionally during their hospital stay.

On the other hand, it is known that intensive care of undernourished patients, including nutritional intervention, has prevented hospital-acquired malnutrition, decreased hospitalization costs, reduced length of hospital stay and the need for readmissions, and improved cognitive, physical, and social functioning, as well as patient/staff satisfaction [20,21,22]. Therefore, it is important to identify subjects malnourished or at risk of malnutrition, since it can influence the hospital course in terms of length of stay, and the prognosis in terms of mortality [21,23,24,25]. 

Serum albumin levels are now considered an important marker in the measurement of malnutrition. Albumin levels are highly predictive of in-hospital mortality [26] and in the general population. For every 2.5 g/L decrease in serum albumin concentration, there is a 24% to 56% increase in the probability of death. Serum albumin levels are indicative of the sum of hepatic synthesis (12–15 g/day), plasma distribution, and protein loss [27,28,29,30,31,32,33,34,35,36,37].

Total lymphocyte count (TLC) is a useful indicator of nutritional status. It is easy to perform, quick, and appropriate for all age groups. TLC decreases with progressive malnutrition and correlates with morbidity and mortality in hospitalized patients [38,39,40].

The INA [41] uses serum albumin and blood lymphocyte counts for nutritional assessment. Patients were classified in four degrees of nutritional state: first degree (serum albumin > 3.5 g/dL; blood lymphocyte count > 1500 cells/mm^3^), second degree (serum albumin > 3.5 g/dL; blood lymphocyte count < 1500 cells/mm^3^), third degree (serum albumin < 3.5 g/dL; blood lymphocyte count > 1500 cells/mm^3^), and fourth degree (serum albumin < 3.5 g/dL; blood lymphocyte count < l500 cells/mm^3^).

The PNI [42,43,44] is calculated using serum albumin and blood lymphocyte count [10 × Albumin (g/dL)] + [0.005 × blood lymphocyte (cells/mm^3^). PNI has been validated as a predictor of mortality in surgical settings. Several trials have shown an association between low PNI values and poor survival in patients affected by several types of cancer [45,46,47,48,49].

Although screening methods including the Universal Malnutrition Screening Tool (MUST), the Mini Nutrition Assessment (MNA) [23,50,51,52,53], the Prognostic Nutritional Index (PNI) [42,43,44], Instant Nutritional Assessment (INA) [54], Body Mass Index (BMI), weight, serum albumin, or blood lymphocyte count) are available, there is no “gold standard”. No single medium exists for screening or assessing nutritional status to predict the above-mentioned poor-nutrition-related outcomes [21,55,56,57,58,59]. 

We conducted an observational, retrospective study on a cohort of patients admitted to an internal and aging medicine department at the “Policlinico Riuniti” University Hospital of Foggia, Italy to assess the ability of the Instant Nutritional Assessment (INA) and the Prognostic Nutritional Index (PNI) to predict hospitalization outcome, i.e., in-hospital mortality and length of stay (LOS). Serum albumin and lymphocytes were evaluated to predict hospitalization outcomes.

## 2. Materials and Methods

### 2.1. Patients

We examined a cohort of 2787 patients admitted to the Internal Medicine and Aging Department of the “Policlinico Riuniti” University Hospital in Foggia, Italy, between 1 January 2019, and 31 December 2022. Study exclusion criteria were age less than 18 years at the time of admission; patients discharged against medical advice; patients transferred to other departments or other hospitals; and patients discharged to nursing homes or rehabilitation institutions. The final cohort consisted of 2206 subjects. 

### 2.2. Methods

Recorded data included C-reactive protein (CRP), INA, PNI, length of stay (LOS), and outcome of hospitalization, i.e., discharge home or death. Serum values of albumin and lymphocytes were recorded from all patients.

We must underline that all patients analyzed in our study were treated with medical therapy. In some cases, the therapy they were already taking at home had been confirmed in whole and in part; in others, the therapy had been modified depending on the clinical situation.

### 2.3. Statistics

After performing the Kolmogorov–Smirnov test and having verified from the test that all the data examined did not follow the normal distribution (*p* < 0.001), the non-parametric Mann–Whitney U-test corrected with the Monte Carlo exact test for the comparison of means for independent samples was performed; also, the non-parametric Spearman test for the calculation of correlations was performed. 

Analysis of the ROC curve was also performed to measure the sensitivity and specificity, or the predictive value of mortality of PNI, as well as to identify the optimal threshold value (best cut-off). We used Cox regression in survival analysis in the prediction of mortality, expressed by the Hazard Ratio (HR), and logistic regression in the prediction of readmission, expressed by the Odds Ratio (OR). Both Cox regression and logistic regression were performed after correction for the age at the time of admission. 

Finally, the Kaplan–Meier analysis was performed to estimate the survival of patients during the observation period, in relation to the examined parameters. The Log-rank test, stratified by age at admission, was performed to compare the two Kaplan–Meier survival curves. 

Statistical analyses were performed using IBM SPSS version 25 (Armonk, NY, USA), and STATA SE 14.2 (College Station, TX, USA), with a significance level of 0.05. 

## 3. Results

The features of the sample examined are presented in Table 1. As expected, women were older than men (*p* < 0.001). No statistically significant differences were observed between men and women for both LOS, serum values of Albumin, PNI, and INA. Men had higher CRP values than women (*p* = 0.013), while women had slightly higher lymphocyte values than men (*p* = 0.043). 

As shown in Table 2, after correcting by sex, the correlation analysis showed, a direct relationship between age at admission, CRP values, and LOS (*p* < 0.001), as expected; an inverse relationship was observed between LOS and PNI (*p* < 0.001). 

Three hundred and twenty-seven patients died during hospitalization. As shown in Table 3, the deceased were older than the not deceased (*p* < 0.001), with no significant differences between males and females (*p* = 1.000), and in terms of LOS (*p* = 0.554). Deceased patients had lower albumin, higher CRP levels, and slightly higher lymphocyte levels (*p* < 0.001) compared to non-deceased patients, resulting in lower PNI values (*p* = 0.001). Again, compared to the non-deceased patients, third- and fourth-degree INA scores were more frequent among the deceased (*p* < 0.001).

The analysis of the ROC curves showed that PNI (Figure 1a), albumin (Figure 1b), lymphocytes (Figure 1c), and CRP (Figure 1d) are significant predictors of mortality. By comparing the Area Under the Curve (AUC), the most accurate predictor of mortality is PNI, which showed an AUC of 0.785, and the best cut-off of 36.6. 

After correcting for age at the time of admission, the Cox regression analysis with the Breslow method was performed. The result of the analysis highlighted that both the PNI value less than 36.6 and the third and fourth-degree INA are risk factors independent of mortality during hospitalization (HR = 3.56, 95% Confidence Intervals = 2.25–5.65, *p* < 0.001; HR = 3.12, 95% Confidence Intervals = 2.35–4.14, *p* < 0.001, respectively). 

We then performed survival analysis using the Log-rank test, stratified by age at admission, to compare the two Kaplan–Meier survival curves. Survival analysis showed that after thirty days from admission, both third- and fourth-degree INA and PNI values under 36.6 were associated with reduced survival (Figure 2a,b).

Finally, for 309 patients, the hospitalization was a rehospitalization, that is, a new hospitalization within thirty days of a previous discharge. Also, for these patients, we considered the INA and the PNI for the evaluation of their nutritional status but viewed them retrospectively. Concerning these patients, a poor nutritional status, expressed both by third and fourth-degree INA and by a PNI value lower than 36.60, constituted a significant risk factor for rehospitalization, as shown in Table 4 (*p* < 0.001).

## 4. Discussion

In our retrospective analysis of a large cohort of hospitalized patients, 327 subjects, or 14.8% of the participants, died. As expected, deceased patients were older than non-deceased patients (*p* < 0.001). The deceased group did not have a significantly longer LOS compared to non-deceased patients (*p* = 0.115); serum albumin and lymphocyte values are indicators of a malnutrition state, which correlate with poor clinical outcomes in patients. Albumin levels were lower, while lymphocyte levels were slightly higher (*p* < 0.001) in deceased patients.

The mortality group had a significantly lower PNI score (*p* = 0.001); also, 93% of the deceased had a third- and fourth-degree INA (*p* < 0.001). 

ROC curve analysis found that the PNI was a significant predictor of mortality outcome with an AUC of 0.785. This agrees with previous work showing that PNI is associated with postoperative mortality in cancer patients [60,61,62,63,64]. 

Our retrospective analysis also showed that both PNI and INA, which can easily be calculated using the patient’s laboratory values, are independent risk factors for mortality during hospitalization (*p* < 0.001), where both third- and fourth-degree INA and PNI values under 36.6 were associated with reduced survival. Therefore, PNI and INA can be used as simple tools for assessing the risk of in-hospital mortality in geriatric patients. 

Finally, third- and fourth-degree INA and PNI values under 36.6 constituted a significant risk factor for rehospitalization (*p* < 0.001).

Our study confirms how albumin is an important measurable indicator of nutritional status and how malnutrition determines low serum lymphocyte values, which contributes to poor clinical outcomes in hospitalized patients [18,65]. Our study also confirms the results of previous studies [7] that have demonstrated the prognostic value that changes in routine blood markers, indicative of both nutritional and inflammatory status, can have. 

Furthermore, this study confirms how the reduced energy and protein intake, which characterizes a large percentage of patients during the first days of hospital admission, can contribute to a higher LOS and an increase in in-hospital mortality.

Our study also confirmed how malnutrition, i.e., the evaluation of nutritional status, should be one of the priorities in hospitalized patients; above all, a priority objective among patients who are already malnourished upon admission to the hospital, or at high risk of malnutrition, is that they receive greater attention to the nutritional aspect.

Previous studies have already described how a negative daily energy and protein balance, especially during the first 5 days of hospitalization and in older patients, i.e., reduced food intake or failure to consume a complete regular meal, is associated with a higher risk of high mortality and 30 days after discharge with a longer LOS [9,66]. It is already known that protein–energy malnutrition strongly contributes to an increased risk of sarcopenia, impaired muscle strength and function, and a worsening of health and immune status, especially in older patients [67]. For elderly patients and those at risk of frailty, a protein intake of 1–1.5 g/kg/day has already been proposed as optimal, i.e., approximately 10–12% of the total caloric intake [68,69].

However, in daily clinical practice, the total protein and calorie intake is almost always lower than the patient’s metabolic needs. These results highlight, as previously reported in the literature, the mandatory need for careful monitoring of the patient’s food intake because hospital undernutrition is an important risk factor for malnutrition [70,71].

The most frequent reason for reduced food intake was not the lack of autonomy in eating, as is commonly thought, but rather the lack of appetite, due in most cases to polypharmacy. An almost equally frequent cause of undernutrition is the prescription of fasting, which in many cases is inappropriate, or even the passage of long periods of fasting during hospitalization [72], such as when waiting for diagnostic tests to be carried out. Fasting times often exceed guideline recommendations, leading to a potential worsening of pre-existing malnutrition conditions or inducing a state of hyper-catabolism in at-risk patients [73]. Artificial nutritional support, even when prescribed promptly, proves insufficient for the prevention of malnutrition in hospitals.

Currently, in most departments, the documentation of nutritional information, both at the time of patient admission and during the hospital stay, is insufficient and approximate. Patients, especially the older ones, are not systematically subjected to evaluation of their nutritional status; data such as body weight, body composition, food intake during hospitalization, and nutritional risk are collected randomly, empirically, and without using validated screening tools [74], which are part of the multidimensional evaluation tools. Because nutrition in hospitalized elderly people is influenced by various factors, a multidisciplinary evaluation intervention would be desirable based on the assumption that nutrition is not a mere hotel aspect, but a key factor of hospital care, like the diagnostic-therapeutic intervention, both to improve protein/energy intake and to limit the risk of adverse outcomes [75].

Our study confirms the importance of evaluating nutritional status at the time of hospitalization; this being a frailty factor significantly correlated with LOS and mortality, especially in older patients.

However, it would be wrong to say that hospitalization itself constitutes a risk factor for the development of malnutrition.

The highest prevalence of malnutrition was indeed observed among older patients, consistent with previous studies [60,76,77,78,79,80]. 

Equally true is that at the time of admission to the hospital, some of the patients reported involuntary weight loss during the previous 6 months, especially among patients with ongoing or newly diagnosed neoplastic diseases, or in general in patients suffering from chronic degenerative diseases. Even the reported loss of appetite before admission, largely related to the presence or new diagnosis of the above pathologies, is undoubtedly responsible for the state of malnutrition [81]. Involuntary pre-hospitalization weight loss is an important prognostic index in the evaluation of nutritional status [82,83].

If anything, it is more correct to reiterate that if malnutrition continues to be underestimated and not adequately treated, or if the evaluation of nutritional status is not among the priorities in the patient’s assessment upon hospital admission, it will constitute an important negative prognostic factor. Poor nutritional status of patients will inevitably produce an increase in negative clinical outcomes, such as in-hospital death and rehospitalization, with a consequent increase in hospital costs [84].

There is currently no consensus on the best method for assessing nutritional status in hospitalized patients. The use of clinical scores such as the PMI and the INA, which are certainly more accurate than the use of a single nutritional parameter, can be a valid tool in identifying the state of malnutrition in hospitalized patients.

There are several limitations of this study. First, this analysis was completed at a single center. Secondly, sociodemographic, lifestyle, and morbidity data were not collected, as well as weight and height, body mass index, education level, smoking, alcohol, time spent watching TV, physical activity in leisure time, and medications taken at home. Finally, we have no information on the patient’s eating habits. Therefore, our study may underestimate the effects of nutrition on mortality in the general population.

Above all, the nutritional assessment was carried out only with PNI and INA, two simple-to-calculate, well-studied, and objective measurements, which have already been highlighted as two good predictors of mortality [41,42,43,44]. It is well known that the outcomes of hospitalization depend on various factors that contribute to determining the state of the fragility of patients, such as the main cause of hospitalization, the presence of underlying chronic pathologies, for example, myocardial infarction, stroke, cancer, septic shock, heart failure; the outcome of the hospitalization also depends on the control of the risks associated with hospitalization, i.e., malnutrition acquired during hospitalization, nosocomial infections, falls, or consequences of prolonged bed rest, such as thromboembolic complications, bedsores, the appearance of depressive symptoms or delirium, as well as the worsening of the already present state of sarcopenia, with a further functional decline compared to the time of hospitalization, and consequent loss of autonomy, self-sufficiency, and further functional improvement. We are aware that all these numerous factors should have been taken into consideration as potential confounding factors regarding the prognosis of geriatric patients.

In addition, our analysis combines all-cause mortality. This could understate and overstate the effects of nutrition and frailty on certain types of diseases in older patients. Further studies with a similar design should be done on various types of diseases and different causes of mortality. To compensate for this, it would be desirable to conduct similar studies in different centers, i.e., also including non-hospitalized elderly populations.

However, while considering the limitations mentioned above, of which we reiterate to be aware, it must be kept in mind that our study aimed to consider two already known parameters, which are easily and quickly calculable, that could indicate the nutritional status at the time of admission independently from other prognostic factors.

Our study, which is retrospective and observational, leads us to another important consideration, namely that the evaluation tools indicated above, which should be determined routinely in daily practice, are not in most cases. This is because the medical staff in most cases ignore its meaning, and the nursing staff interprets the compilation of the rating scales as a further increase in work.

This leads us to reiterate how very important it is to carry out an adequate and complete assessment of the nutritional status of patients at the time of admission, especially the older and more fragile ones. This must be integrated with a careful assessment of the functional status and self-sufficiency, which brings attention to the timely training, which must concern all those who work in contact with geriatric patients.

In any case, the identification of malnourished patients or those at high risk of malnutrition with two tools that are easy to apply upon admission allows for timely planning of the nutritional intervention. This will certainly bring benefits to the patient both in terms of functional status and prognosis and in terms of reduction of adverse outcomes such as in-hospital death or readmission within thirty days of discharge. 

Further specifically designed, prospective, and multicentric studies are desirable to correlate the patient’s nutritional status with chronic pathologies and the state of functional autonomy at the time of hospitalization in an acute setting, the presence of complications during hospitalization, and the outcomes of that hospitalization (length of hospitalization, death, or discharge to home or rehabilitation facilities or nursing homes) using simple and rapid administration tools.

## 5. Conclusions

In this large retrospective study, PNI and INA are two simple and quick-to-calculate tools that can help classify the condition of hospitalized elderly patients based on their nutritional status, or in assessing their mortality risk. Furthermore, poor nutritional status at the time of discharge may represent an important risk factor for rehospitalization in the following thirty days. This work highlights the need for future multicenter studies specifically designed on larger cohorts of patients, which also take into account autonomy, chronic poly pathologies, mobility, the risk of falls, and the risk of developing pressure ulcers, i.e., those which take into account the patient’s level of frailty, both to confirm the predictive power of nutritional scores in determining patient outcomes, calculated at the time of admission, and to evaluate the effect of any interventions, including those aimed at improving nutritional status. We strongly reiterate the need for adequate training of doctors and nurses regarding the importance of the nutritional status of patients, i.e., that maintaining a good nutritional status is an integral part of the therapeutic process of hospitalization in acute departments and must not be understood as simple hotel service. We also suggest that upon admission to the hospital, the recording of the quantity of food consumed be included among the daily monitoring parameters, especially in malnourished patients or patients at risk of malnutrition, which also includes the percentage calculation of protein intake compared to the total energy income. In a subsequent study, it would also be interesting to stratify the cohort on specific causes of mortality (sepsis, heart disease, etc.). This would be useful in delineating which patients would benefit most from each specific nutritional intervention.

## Figures and Tables

**Figure 1 nutrients-16-01359-f001:**
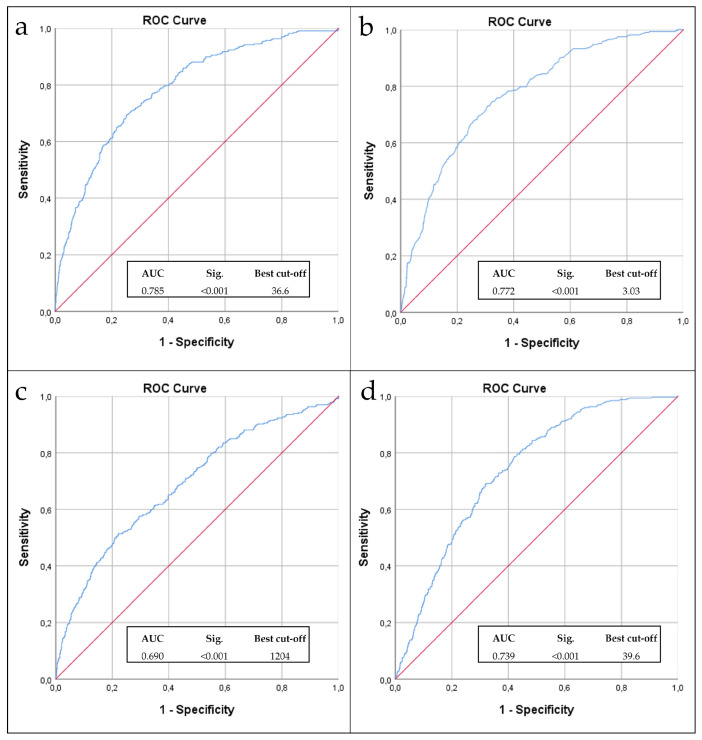
Analysis of ROC curves of PNI, Albumin, Lymphocytes, and CRP as predictors of mortality, compared with the reference line (red line).

**Figure 2 nutrients-16-01359-f002:**
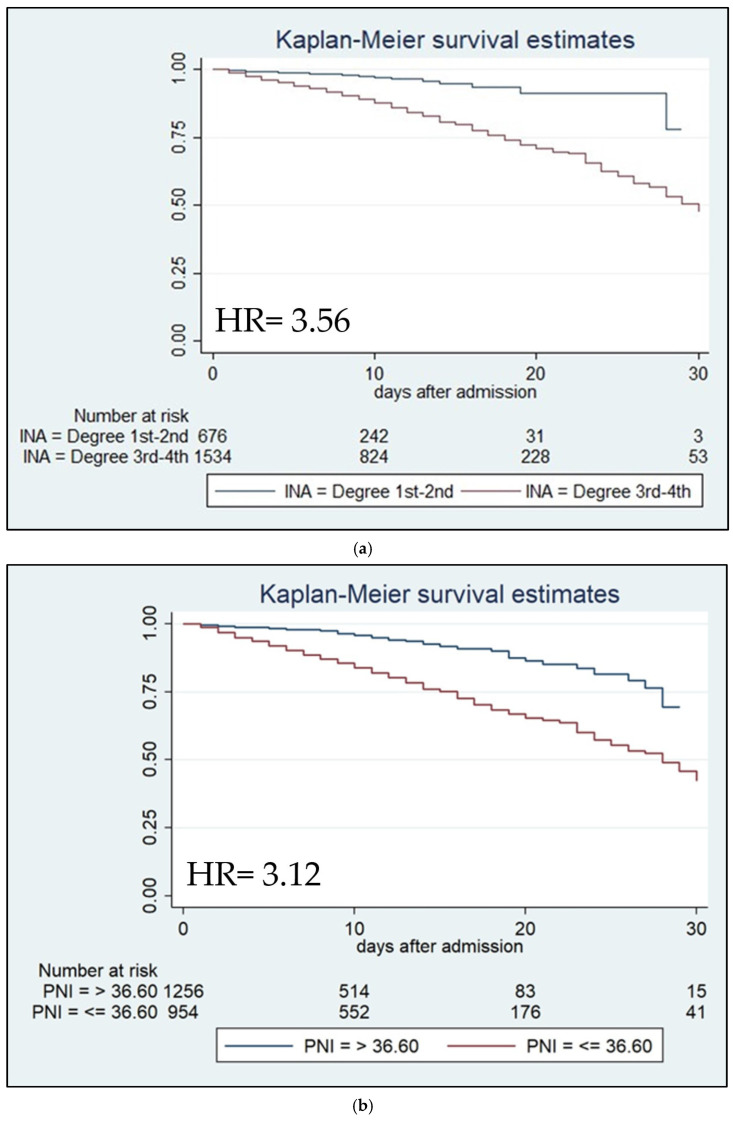
(**a**). Kaplan–Meier survival estimates stratified by third- and fourth-degree vs. first- and second-degree INA (chi-square 33.42; *p* < 0.001). (**b**). Kaplan–Meier survival estimates stratified by ≤36.60 vs. >36.60 PNI value (chi-square 69.45; *p* < 0.001).

**Table 1 nutrients-16-01359-t001:** Clinical features of patients, stratified by sex. Values are expressed as number (%) or as mean ± SD.

	Male	Female	Sig.
**Subjects N (%)**	1112 (50.3%)	1100 (49.7%)	1.000
**Age at hospitalization (Mean ± SD)**	71 ± 15	76 ± 15	<0.001
**LOS (Mean ± SD)**	11 ± 7	11 ± 7	0.242
**Albuminemia (Mean ± SD)**	3.1 ± 0.7	3.1 ± 0.7	0.237
**CRP (Mean ± SD)**	71.8 ± 87	63.2 ± 81	0.013
**Lymphocytes (Mean ± SD)**	1455 ± 1754	1615 ± 4108	0.043
**PNI (Mean ± SD)**	38.7 ± 11.7	39.4 ± 25.5	0.914
**INA score 1st–2nd degree N (%)**	354 (52.2%)	324 (47.8%)	0.321
**INA score 3rd–4th degree N (%)**	758 (49.4%)	776 (50.6%)	

**Table 2 nutrients-16-01359-t002:** Correlation between LOS and age at admission, serum albumin, lymphocytes, and PNI after correcting by sex.

		Length of Stay
**Age at admission**	Correlation	0.094
	Significance (2-tailed)	<0.001
**CRP**	Correlation	0.178
	Significance (2-tailed)	<0.001
**PNI**	Correlation	−0.075
	Significance (2-tailed)	<0.001

**Table 3 nutrients-16-01359-t003:** Clinical features of patients, stratified by deceased and not deceased. Values are expressed as number (%) or as mean ± SD.

	Deceased	Not Deceased	Sig.
**Subjects N (%)**	327 (14.8%)	1885 (85.2%)	<0.001
**Male N (%)**	164 (50.2%)	948 (50.3%)	1
**Female N (%)**	163 (49.8%)	937 (49.7%)	
**Age at hospitalization (Mean ± SD)**	81 ± 11	73 ± 15	<0.001
**LOS (Mean ± SD)**	12 ± 10	11 ± 7	0.554
**Albuminemia (Mean ± SD)**	2.6 ± 0.6	3.2 ± 0.7	<0.001
**Lymphocytes (Mean ± SD)**	1558 ± 6999	1530 ± 1786	<0.001
**PNI (Mean ± SD)**	33.4 ± 35.5	40.0 ± 15.3	<0.001
**CRP (Mean ± SD)**	120.1 ± 95.4	58.4 ± 78.9	<0.001
**INA score 1st–2nd degree N (%)**	23 (7.0%)	655 (34.7%)	<0.001
**INA score 3rd–4th degree N (%)**	304 (93.0%)	1230 (65.3%)	

**Table 4 nutrients-16-01359-t004:** Relationship between third- and fourth-degree INA and PNI values and rehospitalization, after correcting by age at admission.

	**Odds Ratio**	**95% Confidence Intervals**	**Sig.**
**INA Score 3rd–4th degree**	3.14	2.22–4.46	<0.001
**PNI < 36.6**	2.39	1.85–3.10	

## Data Availability

Data is unavailable due to privacy restrictions.

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
