# Peer review of "Prognostic Nutritional Index and Instant Nutritional Assessement Are Associated with Clinical Outcomes in a Geriatric Cohort of Acutely Inpatients"

_nutrients, 2024, doi:10.3390/nu16091359_

Round 1
Reviewer 1 Report
Comments and Suggestions for Authors
In manuscript titled: “Prognostic Nutritional Index and Instant Nutritional Assessment are associated with clinical outcomes in a cohort of hospitalized patients” the authors in retrospective analysis confirmed previously well documented a prognostic value of two clinical scores (PNI and INA) in prediction of such outcomes measured as: length of stay (LOS), mortality, and risk of rehospitalization. Authors justly wrote (page8, line 232) that blood albumin concentration and lymphocytes count, which are the main parameters for PNI and INA calculation, are recognized not only as indices of nutritional risk and nutritional status assessment, but also as biomarkers of patient’s inflammatory status. However, the authors interpreted finally PNI value and INA degree only as parameters of nutritional status assessment, and only with adjustment for patient’s age. Moreover, it is known, that inpatients outcomes depends mainly on: main cause of hospitalization (basic disease, e.g. myocardial infarction, stroke, cancer, septic shock, cardiac failure), comorbidities (e.g. Charlsson comorbidity index), and the control of the risks associated with hospitalization, which are: hospitalization acquired malnutrition (expressed, e.g. by NRS2002 score ≥ 3, and ≥5); hospitalization acquired infection (HAI, e.g. ATLAS score); thromboembolic complications (e.g. Padua and Caprini prediction scores), falls (e.g. MFS score); bedsore (pressure wound, Norton’s scale); occurrence of depressive disorders, delirium, and alienation; as well as immobilization leading to hospital acquired sarcopenia and functional impairment. Geriatric inpatient’s prognosis depends also on their functional status at admission (e.g. Barthel, IADL, and ADL scores) and frailty score. These numerous factors, which are routinely determined in majority of European hospitals, should be taken into consideration, for example as potential confounding factors, in every analysis concerning geriatric patient’s prognosis. Unfortunately, authors did not provide any data concerning factors enumerated above, as well as biomarkers of inflammatory response which certainly determined blood albumin concentration and lymphocytes count. Therefore, in my opinion, this manuscript cannot be considered as suitable for publication, because of it provides none of news to actual knowledge and analysis presented is potentially biased.
Minor remarks:
- Full stop should be removed from the end of title
- Title should contain information that manuscript concern geriatric patients admitted in urgent mode (e.g. “… in a geriatric cohort of acutely inpatients”)
- Frail is not the same as malnutrition, cachexia, sarcopenia, and does not concern all geriatric patients
In METHODS section
- authors did not provide information concerning obtaining method of data concerning rehospitalization (by telephone interview, medical documentation)
- criteria for INA degrees contain “o”, e.g. o1500 cells? (line 86 and 87)
- it is true (line 93) that PNI was biomarker of poor survival in cancer patients, but all publications cited concerned operated patients, and we can only suspect, that geriatric patients analyzed in revised manuscript were treated medically;
- authors should divide this section in subsections, e.g. Patients, Methods, Statistic, Bioethic
- test was used for determination of variables distribution, why authors used both parametric and non-parametric tests?
Results section:
- Titles of tables should be given above not under the Table
- What was the rationale for comparison male and female patients in Table 1?
- Table 2 and line 212- the Spearman correlation between LOS and parameters determined, e.g. blood albumin concentration had very low values of R-coefficient, and provide no important information to the manuscript content
- In main opinion analysis related to manuscript title begins from Table 3
- Why authors did not perform separate ROC curve analysis in regard to every outcome measured? PNI value cut-off for in-hospital mortality was the same as for readmission?
- Figure 1 require improvement- Table with AUC should be presented completely, and should contain cutoffs
- Table 4 can be removed
- In Tables comas should be replaced by dots
- Why Table 5 presents HR and Table 6 OR? This is not the same
- In Figures 2a and 2b - I suggest to limit observation duration to 20 days, maximally to 40 days, because the Kaplan Meier lines course is biased by statistical outliers, the mean ± SD for LOS amounted to 11±7 (Table 1)
- Which test was used for comparing two Kaplan-Meier survival curves (Breslow test, Cox’s test, log-rank)? Which was statistical significance?
DISCUSSION SECTION
- Line 231 it is difficult to say that lymphocytes count and albumin blood concentration are routine blood markers, even in geriatric patients
- Frailty severity was not determined in this study (line 242)
CONCLUSIONS
Not all conclusion are based on results obtained, for example: (line 249-250): In this large retrospective study, PNI and INA are two simple and rapid-to-calculate tools that can better classify frail and elderly patients regarding their nutritional status and possible clinical outcomes – better compared to what? Frail was not determined, and “frail” is not synonym to “geriatric”
REFERENCES
- Reference 6- updated European consensus on sarcopenia definition and diagnosis is available
- Reference 16 is duplicated as 31 and 32
- References 43 and 48, 44 and 49, 45 and 50, 46 and 51, 47 and 52 are duplicated
Author Response
Comment 1: In manuscript titled: “Prognostic Nutritional Index and Instant Nutritional Assessment are associated with clinical outcomes in a cohort of hospitalized patients” the authors in retrospective analysis confirmed previously well documented a prognostic value of two clinical scores (PNI and INA) in prediction of such outcomes measured as: length of stay (LOS), mortality, and risk of rehospitalization. Authors justly wrote (page8, line 232) that blood albumin concentration and lymphocytes count, which are the main parameters for PNI and INA calculation, are recognized not only as indices of nutritional risk and nutritional status assessment, but also as biomarkers of patient’s inflammatory status. However, the authors interpreted finally PNI value and INA degree only as parameters of nutritional status assessment, and only with adjustment for patient’s age. Moreover, it is known, that inpatients outcomes depends mainly on: main cause of hospitalization (basic disease, e.g. myocardial infarction, stroke, cancer, septic shock, cardiac failure), comorbidities (e.g. Charlsson comorbidity index), and the control of the risks associated with hospitalization, which are: hospitalization acquired malnutrition (expressed, e.g. by NRS2002 score ≥ 3, and ≥5); hospitalization acquired infection (HAI, e.g. ATLAS score); thromboembolic complications (e.g. Padua and Caprini prediction scores), falls (e.g. MFS score); bedsore (pressure wound, Norton’s scale); occurrence of depressive disorders, delirium, and alienation; as well as immobilization leading to hospital acquired sarcopenia and functional impairment. Geriatric inpatient’s prognosis depends also on their functional status at admission (e.g. Barthel, IADL, and ADL scores) and frailty score. These numerous factors, which are routinely determined in majority of European hospitals, should be taken into consideration, for example as potential confounding factors, in every analysis concerning geriatric patient’s prognosis. Unfortunately, authors did not provide any data concerning factors enumerated above, as well as biomarkers of inflammatory response which certainly determined blood albumin concentration and lymphocytes count. Therefore, in my opinion, this manuscript cannot be considered as suitable for publication, because of it provides none of news to actual knowledge and analysis presented is potentially biased.
Response 1: Thank you very much for taking the time to review this manuscript. We agree with your comments. We indeed considered the PNI value and the INA grade as the only parameters for evaluating the nutritional status and only with adjustment for the patient's age for the evaluation of the outcomes. it is also true that a major limitation of our study was not having considered both the diagnosis at admission to hospitalization, the comorbidities (for example Charlsson comorbidity index), and the evaluation of the risks associated with hospitalization, i.e. malnutrition acquired during hospitalization, infections acquired during hospitalization, thromboembolic complications, bedsores, the appearance of depressive disorders, or delirium. We agree with you that not considering all these potential confounding factors could lead to biased results. However, the truth is that most of the conditions, especially malnutrition and sarcopenia, are the consequence of chronic conditions that are most often already present at the time of hospitalization. If anything, the presence of complications during hospitalization in an acute setting, to be correlated with the outcomes of hospitalization (length of hospitalization, death, or discharge to home or rehabilitation facilities or to nursing homes, could be the subject of a subsequent study, specifically designed, prospective and also multicentre. The aim of our study was in fact to consider two parameters already known, easily and quickly calculable, which could indicate the nutritional status at the time of hospitalisation. We also agree with your observation that the prognosis of hospitalized geriatric patients also depends on their functional status at the time of admission, detected by the Barthel, IADL, and ADL, and by the frailty score. In reality, our study did not only include geriatric patients, but your correct observation leads to an important consideration, that is, the evaluation tools you indicated, which should be determined routinely, in practice in most cases are not; this is because the medical staff in most cases ignores its meaning, and the nursing staff interprets the compilation of the evaluation scales as a further increase in work; the story returns to the theme addressed several times in the literature of training in the care of geriatric patients in acute wards.
Minor remarks:
Comment 2: Full stop should be removed from the end of title
Response 2: As you suggested, the full stop has been removed from the end of the title.
Comment 3: Title should contain information that manuscript concern geriatric patients admitted in urgent mode (e.g. “… in a geriatric cohort of acutely inpatients”)
Response 3: Thank you for pointing this out. As you suggested, the period has changed as follows: "Prognostic Nutritional Index and Instant Nutritional Assessement are associated with clinical outcomes in a geriatric cohort of acutely inpatients".
Comment 4: Frail is not the same as malnutrition, cachexia, sarcopenia, and does not concern all geriatric patients
Response 4: Thank you very much for this observation. We agree with this comment. We have therefore modified the text on line 38 by inserting what we believe is the most correct definition of fragility, naturally inserting the references, as follows: "The geriatric patient is not characterized solely by advanced age, but rather by the presence of numerous chronic pathologies, together with the presence in many cases of the well-known geriatric syndrome which is frailty. By frailty, we mean the state of greater vulnerability, or reduced resilience, in response to a stressful event, which increases the risk of adverse outcomes, including falls, delirium, and disability [8-10 3,5,6]. In these subjects, a small insult (for example a new drug, a small infection, or a small surgery) causes a significant and disproportionate change in the state of health, i.e. from independent to dependent, from mobile to immobile, from postural stability to propensity for falling, or from lucid to delirious."
References:
Fried LP, Tangen CM, Walston J, Newman AB, Hirsch C, Gottdiener J, Seeman T, Tracy R, Kop WJ, Burke G, McBurnie MA. Frailty in older adults: evidence for a phenotype. J Gerontol A Biol Sci Med Sci. 2001;56(3):M146–56.
Walston J, Hadley EC, Ferrucci L, Guralnik JM, Newman AB, Studenski SA, Ershler WB, Harris T, Fried LP. Research agenda for frailty in older adults: toward a better understanding of physiology and etiology: summary from the American Geriatrics Society/National Institute on Aging Research Conference on Frailty in Older Adults. J Am Geriatr Soc. 2006;54(6):991–1001.
Eeles EM, White SV, O’Mahony SM, Bayer AJ, Hubbard RE. The impact of frailty and delirium on mortality in older inpatients. Age Ageing. 2012;41(3):412–6.
Clegg A, Young J, Iliffe S, Rikkert MO, Rockwood K. Frailty in elderly people. Lancet. 2013; 381(9868): 752-62.
In METHODS section
Comment 5: Authors did not provide information concerning obtaining method of data concerning rehospitalization (by telephone interview, medical documentation)
Response 5: Thank you for pointing that out. We obtained data relating to rehospitalization from the medical record. We corrected the concept that these were retrospective data, specifying that for some cases the subjects' current hospitalization was a rehospitalization from a recent hospitalization.
Comment 6: Criteria for INA degrees contain “o”, e.g. o1500 cells? (line 86 and 87)
Response 6: Thank you for pointing this out. We corrected: “blood lymphocyte count 1500 cells/mm3”.
Comment 7: It is true (line 93) that PNI was biomarker of poor survival in cancer patients, but all publications cited concerned operated patients, and we can only suspect, that geriatric patients analyzed in revised manuscript were treated medically;
Response 7: Thank you for your observation. We agree with your comment. Therefore, we underlined in the text that all patients analyzed in our study were treated with medical therapy; in some cases, the therapy they were already taking at home had been confirmed in whole and in part, in others the therapy had been modified depending on the clinical situation.
Comment 8: Authors should divide this section in subsections, e.g. Patients, Methods, Statistic, Bioethic
Response 8: Thank you very much. I followed your suggestion.
Comment 9: Test was used for determination of variables distribution, why authors used both parametric and non-parametric tests?
Response 9: Thank you for your observation. We performed the Kolmogorov-Smirnov test; after verifying that all the data examined did not follow the normal distribution (p<0.001), we performed the non-parametric Mann-Whitney U test corrected with the Monte Carlo exact test for the comparison of means for independent samples, and the test did not - Spearman's parametric test was performed to calculate the correlations. We have therefore corrected the text and tables.
In RESULTS section:
Comment 10: Titles of tables should be given above not under the Table
Response 10: We gave the titles above the table, as you suggested.
Comment 11: What was the rationale for comparison male and female patients in Table 1?
Response 11: Our rationale for comparing male and female patients is derived from the evidence that there is substantial variability in the aging process between men and women. In general, women live longer than men, but are more frail in old age, resulting in poorer health. On the contrary, men, despite having a reduced life expectancy compared to women, maintain good levels of physical functionality even at an advanced age, resulting in better health and a lower risk of frailty. In this regard, we have inserted a correction in the text with the relevant reference.
Reference: Hagg S and Jylhava J. Sex differences in biological aging with a focus on human studies eLife 2021;10:e63425
Comment 12: Table 2 and line 212- the Spearman correlation between LOS and parameters determined, e.g. blood albumin concentration had very low values of R-coefficient, and provide no important information to the manuscript content
Response 12: Thank you very much for your observation. I wanted to include the albumin levels and lymphocytes in the Spearman correlation as they are the parameters for calculating the PNI. We agree with you both that these variables have very low values of r-coefficient and that they do not provide important information on the content of the manuscript; therefore we removed them from the analysis, which now indicates the correlation between length of stay, age at admission, c-reactive protein and PNI.
Comment 13: In main opinion analysis related to manuscript title begins from Table 3
Response 13: Thank you very much. We have reorganized the description of the results by focusing attention from Table 3 onwards. Tables 1 and 2 have a merely descriptive function of the characteristics of the population under consideration.
Comment 14: Why authors did not perform separate ROC curve analysis in regard to every outcome measured? PNI value cut-off for in-hospital mortality was the same as for readmission?
Response 14: Thank you very much for your suggestion. We performed four separate ROC curve analyses regarding PNI, albumin, lymphocytes, and CRP, denoted as Figures 1a, 1b, 1c, and 1d, respectively. Concerning the cut-off of the PNI value for hospital mortality, we have considered the same value for readmission.
Comment 15: Figure 1 require improvement- Table with AUC should be presented completely, and should contain cutoffs.
Response 15: We have presented a separate table, indicated as Table 4, separate from figures 1a, 1b, 1c, and 1d, with the AUC values and best cut-offs.
Comment 16: Table 4 can be removed
Response 16: Table 4 has been removed
Comment 17: In Tables comas should be replaced by dots
Response 17: We replace all commas with dots in Tables
Comment 18: Why Table 5 presents HR and Table 6 OR? This is not the same
Response 18: Thanks for pointing that out. Regarding Table 5, we wanted to consider both INA and PNI as risk factors for mortality during hospitalization, i.e. viewed prospectively, for which we used the HR.
As regards Table 6, we wanted to consider the nutritional status, calculated with INA and PNI, of patients for whom hospitalization was a rehospitalization. That is, we considered the same parameters, with the same PNI cut-off considered to evaluate the risk of mortality, viewed retrospectively, for which we used the OR.
Comment 19: In Figures 2a and 2b - I suggest to limit observation duration to 20 days, maximally to 40 days, because the Kaplan Meier lines course is biased by statistical outliers, the mean ± SD for LOS amounted to 11±7 (Table 1)
Response 19: Thank you very much. We followed your suggestion and modified figures 2a and 2b
DISCUSSION SECTION
Comment 20: Line 231 it is difficult to say that lymphocytes count and albumin blood concentration are routine blood markers, even in geriatric patients
Response 20: Thank you for pointing this out. To have a more stable value, or to reduce potential bias, we calculated albumin indirectly starting from the serum total protein values and the percentage value reported on serum protein electrophoresis, commonly present in our routine blood tests, as follows: albumin = (total protein x albumin%)/100. We applied a similar formula for the lymphocyte count, starting from the white blood cell count and the percentage values of the leukocyte formula of the complete blood count, as follows: lymphocytes = (white blood cells x lymphocytes%)/100.
Comment 21: Frailty severity was not determined in this study (line 242)
Response 21: Thanks for pointing that out. Frailty severity of the patients was not determined because it was not within the scope of the study. The main aim of the study was to determine the nutritional status of patients at the time of admission to the ward and correlate it with the length of hospitalization and the risk of death, or to establish the state of malnutrition as a risk factor for rehospitalization. However, the severity of the frailty at the time of admission to the ward together with the evaluation of the nutritional status can be an interesting starting point for a subsequent study.
CONCLUSIONS
Comment 22: Not all conclusion are based on results obtained, for example: (line 249-250): In this large retrospective study, PNI and INA are two simple and rapid-to-calculate tools that can better classify frail and elderly patients regarding their nutritional status and possible clinical outcomes – better compared to what? Frail was not determined, and “frail” is not synonym to “geriatric”
Response 22: Thank you very much for pointing this out. We agree with this comment. Therefore, we have completely revised the conclusions. in particular, we changed the beginning of the conclusions as follows:
"In this large retrospective study, PNI and INA are two simple and quick-to-calculate tools that can help in classifying the condition of hospitalized elderly patients also based on their nutritional status, or in assessing their mortality risk."
REFERENCES
Comment 23: Reference 6- updated European consensus on sarcopenia definition and diagnosis is available
Response 23: Thank you. I have updated the reference above.
Comment 24: Reference 16 is duplicated as 31 and 32
Response 24: Thank you. I deleted the duplicate reference and corrected the in-text citation.
Comment 25: References 43 and 48, 44 and 49, 45 and 50, 46 and 51, 47 and 52 are duplicated
Response 25: Thank you. I deleted the duplicate references and corrected the in-text citations.
Reviewer 2 Report
Comments and Suggestions for Authors
I read with great interest the manuscript of Capurso and colleagues. In this manuscript two valid scores are associated with nutritional status and clinical outcomes in older, which I found very interesting. However, I have several comments for the authors:
- Please, soften your message since this is a retrospective observational study. Use de formulas "may be" or "may be associated" for your conclusions.
- Please, explain briefly why serum albumin and lymphocytes were also evaluated (I guess they were evaluated since this variables are nutritional related).
- Advice: write a table with PNI and INE variables.
- Statistical methods need a subheading.
- Could the authors provide more data about the clinical condition of the included patients?
- Could the authors develop a little bit the discussion?
- Please, include the retrospective observational nature of your study as a limitation and write all the limitations in a single paragraph.
- Please, be cautious with your conclusions and soften the message itself.
Minor comments:
- Lines 84-89. Please, review the writting the degrees of Lymphocyte count and albumin.
- Lines 91-94. This comment correpond to discussion.
- Kaplan Meier figures need the meaning in the axis or explanation in the legend.
Author Response
Comment 1: Please, soften your message since this is a retrospective observational study. Use de formulas "may be" or "may be associated" for your conclusions.
Response 1: Thank you very much for your observation. I followed your suggestion
Comment 2: Please, explain briefly why serum albumin and lymphocytes were also evaluated (I guess they were evaluated since this variables are nutritional related).
Response 2: Thank you for your suggestion. I have included two paragraphs in the Introduction that explain the importance of serum albumin values and total lymphocyte counts in the evaluation of malnutrition.
Comment 3: Advice: write a table with PNI and INA variables.
Response 3: Thank you very much for your suggestion. I have corrected the paragraphs in the Introduction that explain the importance of the PNI and INA in the evaluation of malnutrition. The calculation of the PNI and INA variables is reported in the text and is reported in all tables of the manuscript.
Comment 4: Statistical methods need a subheading.
Response 4: Thank you very much. I followed your suggestion
Comment 5: Could the authors provide more data about the clinical condition of the included patients?
Response 5: Thank you very much for your observation. Unfortunately, this is a strong limitation of this study, due to the retrospective nature, as we reported in the discussion. Sociodemographic, lifestyle and previous morbidity data were not collected for analysis, as were weight and height, body mass index or education level, smoking, alcohol, time spent watching TV, leisure time physical activity, and medications taken at home. Finally, we have no information on the patient's eating habits. We are aware that our study may underestimate the effects of nutrition on mortality in the general population.
Comment 6: Could the authors develop a little bit the discussion?
Response 6: Thank you very much for your suggestion. I revised the conclusions extensively.
Comment 7: Please, include the retrospective observational nature of your study as a limitation and write all the limitations in a single paragraph.
Response 7: Thank you very much for your suggestion. I followed your suggestion and I revised the conclusions extensively.
Comment 8: Please, be cautious with your conclusions and soften the message itself.
Response 8: Thank you very much. I followed your suggestion
Minor comments:
Comment 9: Lines 84-89. Please, review the writting the degrees of Lymphocyte count and albumin.
Response 9: Thank you very much for your observation. I have corrected the degrees of Lymphocyte count and albumin, as you suggested.
Comment 10: Lines 91-94. This comment correpond to discussion.
Response 10: Thank you very much for your observation. I moved the lines 85 - 94 in the introduction. and revised the conclusions extensively.
Comment 11: Kaplan Meier figures need the meaning in the axis or explanation in the legend.
Response 11: Thank you very much. I have corrected Kaplan Meier figures, as you suggested.
Round 2
Reviewer 1 Report
Comments and Suggestions for Authors
Dear authors! Thank you for taking my suggestions into account
Minor revisions:
- I suggest to joint Figures 1a-1d in one Figure, moreover AUC values given in Table 4 can be presented in respective Figures, and then Table 4 can be removed
- HRs given in Table 5 can be presented on respective figures 2a and 2b, and then Table 5 can be removed
- I still cannot understand why authors use HR in prediction mortality (Table 5), and OR in prediction of readmission (Table 6). The only explanation is that they used Cox regression in survival analysis in prediction of mortality, and logistic regression in prediction of readmission, however they did not explain this in statistics section.
- I still cannot understand why authors cite so many papers published in past century when more actual publications are available
Author Response
Comment 1: I suggest to joint Figures 1a-1d in one Figure, moreover AUC values given in Table 4 can be presented in respective Figures, and then Table 4 can be removed
Response 1: Thank you very much. I followed your suggestion.
Comment 2: HRs given in Table 5 can be presented on respective figures 2a and 2b, and then Table 5 can be removed
Response 2: Thank you very much. I followed your suggestion.
Comment 3: I still cannot understand why authors use HR in prediction mortality (Table 5), and OR in prediction of readmission (Table 6). The only explanation is that they used Cox regression in survival analysis in prediction of mortality, and logistic regression in prediction of readmission, however they did not explain this in statistics section.
Response 3: Thank you for pointing this out. We agree with this comment. We used Cox regression in survival analysis in the prediction of mortality, and logistic regression in the prediction of readmission. We followed your suggestion and we explained this in the statistics section.
Comment 4: I still cannot understand why authors cite so many papers published in past century when more actual publications are available.
Response 4: Thank you for pointing this out. We agree with this comment. We performed an in-depth literature search. There are indeed several more recent references in the literature, but we believe that the older ones, which are then cited in the references, were more in keeping with the hypothesis of our work.
Reviewer 2 Report
Comments and Suggestions for Authors
Dear authors,
I have reviewed with great interest the present manuscript based on my previous concerns. From my point of view, this manuscript would have scientific background. I think authors have achieved and answered all, with great improvement through all the manuscript. In consequence, no further changes are needed along the manuscript.
Author Response
Thank you very much. However, we performed a further revision of the manuscript, following the further suggestions of reviewer 1.